# Pilot and Feasibility Studies of a Lifestyle Modification Program Based on the Health Belief Model to Prevent the Lifestyle-Related Diseases in Patients with Mental Illness

**DOI:** 10.3390/healthcare11121690

**Published:** 2023-06-08

**Authors:** Naomi Tsubata, Akiko Kuroki, Harumi Tsujimura, Masako Takamasu, Nariaki IIjima, Takashi Okamoto

**Affiliations:** 1Junwakai Yahagigawa Hospital, 141 Minamiyama, Fujii-cho, Anjo 448-0023, Japan; 2Department of Home Economics, Faculty of Home Economics, Japan Women’s University, 2-8-1 Mejirodai, Bunkyo-ku, Tokyo 112-0015, Japan; 3Department of Molecular and Cellular Biology, Nagoya City University Graduate School of Medical Sciences, 1 Kawasumi, Mizuho-cho, Mizuho-ku, Nagoya 467-86-1, Japan; 4Division of Internal Medicine, Yahagigawa Hospital, Anjo 448-0023, Japan

**Keywords:** health belief model (HBM), lifestyle-related diseases, schizophrenia (SZ), bipolar disorder (BD), metabolic syndrome (MetS)

## Abstract

In this study we have examined the feasibility of a program based on the health belief model (HBM), for its effectiveness in improving lifestyle-related diseases in patients with schizophrenia (SZ) and bipolar disorder (BD), which are often complicated with physical conditions. In this model, we attempted to enable patients to identify a “threat” and to find “balance between benefits and disadvantages”. Subjects were carefully selected from among psychiatric patients by excluding any bias. Thus, the enrolled patients were 30 adult men and women with lifestyle-related diseases, or those with a body mass index (BMI) of over 24. Of these 30 subjects, 15 were randomly assigned to the intervention group and 10 the control group, since 5 subjects in the control voluntarily left from the study. Comparison of the intervention and control groups revealed significant improvement (*p* < 0.05) in HDL cholesterol in the intervention group. However, there were no significant changes in other variables. These findings support the usefulness and efficacy of HMB-based nutritional interventions for preventing lifestyle-related disorders among psychiatric patients. Further evaluation is needed with a larger sample size and a longer intervention period. This HMB-based intervention could be useful for the general population as well.

## 1. Introduction

### 1.1. Background

Lifestyle-related diseases is a generic term for diseases such as diabetes and obesity, which are strongly related to lifestyle factors such as diet, exercise, rest, smoking, and alcohol consumption. They are leading causes of death among people throughout the world [1]. In addition, the total number of patients with mental disorders in Japan was approximately 5,032,000 as of 2020, according to a survey carried out in Japan by the Ministry of Health, Labor and Human Welfare [2].

A relationship between lifestyle-related illnesses and mental disorders has long been observed. For instance, it was suggested that patients with schizophrenia (SZ) and bipolar disorder (BD) have a higher risk of obesity, diabetes, and dyslipidemia than the general population [3,4,5,6]. Strassnig et al. [7] reported that 62% of SZ and 50% of BD patients were obese 20 years after their first hospitalization for psychosis. The proportions were significantly higher than in the general adult population, where only 27% reached the level of obesity in 20 years. This indicates that the probability of developing obesity and lifestyle-related diseases is high in SZ and BD patients.

The higher prevalence of obesity in SZ and BD patients is considered ascribable to the difficulty in self-management because of low levels of understanding and cognition, due to impaired mental function [8,9]. In addition, these patients often experience tobacco use, lowered physical activity, and higher caloric intake, eventually leading to obesity and hyperlipidemia [10,11]. Moreover, it is well known that some antipsychotic medications often contribute to lifestyle-related diseases and weight gain in patients with psychiatric disorders [12,13,14,15,16,17]. These findings suggest that patients with SZ and BD are likely to develop lifestyle-related diseases as described earlier, which has to be substantiated through anthropometric and biochemical parameters [18,19]. Thus, the practical efficacy of such educational intervention in patients with psychotic disorders is examined in this study.

### 1.2. Objective

The main objective of this pilot study is to implement a lifestyle modification program based on the health belief model (HBM), for patients with SZ or BD and lifestyle-related diseases or a BMI of 24 or higher, and to verify the efficacy of the program based on BMI values and relevant biochemical data.

The HBM is a social model developed by Becker et al. [20], which attempts to change behavior by informing subjects of the personal threat of disease and also developing confidence in the effectiveness of recommended health behaviors, and eventually leading to meaningful action toward such health-promoting behaviors. The HBM has also been used in nutrition education to promote healthy eating behaviors [21,22], as well as in a wide range of other areas such as exercise, diabetes management, and health-promoting behaviors [23,24,25]. Previous studies have reported that diet and exercise programs for patients with schizophrenia can lead to significant improvements in BMI, exercise habits, HbA1c levels, blood pressure, and nutritional knowledge [26,27,28,29,30]. However, to our knowledge, no studies on patients with mental illnesses have been attempted that have implemented an intervention program based on the health belief model.

This study has attempted to examine the efficacy of an HBM-based intervention program for reducing the risk of lifestyle-related illness in psychiatric patients. 

## 2. Methods

### 2.1. Subjects

Adult male and female patients with psychiatric disorders such as SZ and BD, who were hospitalized at the Yahagigawa Hospital, Anjo, Japan or attending its outpatient rehabilitation care center, were selected. Subjects were either experiencing lifestyle-related diseases (diabetes, dyslipidemia, hypertension, or hyperuricemia) or had a BMI of 24 or higher, and gave consent to this study. Patients in acute care wards who were likely to be discharged from the hospital during the study period, and those who attended the outpatient rehabilitation irregularly, were excluded. These subjects were not assigned to the study. They did not take part in other clinical studies.

### 2.2. Study Design

Subjects were randomly assigned to the intervention or the control groups. The eventual sizes were different because some patients left this study due to the lack of nutritional intervention. For the intervention group, a total of eight sessions of a lifestyle improvement program (henceforth referred to as the “program”) was conducted, consisting of a lecture, group work, review of goal setting, and individual consultation time. Before this study had begun (before program implementation) and after program completion (or after 4 months), both of these subject groups were surveyed with regard to their lifestyle and knowledge of appropriate lifestyle habits.

Program implementation and the survey were conducted at Yahagigawa Hospital from September 2018 to January 2019. Lifestyle surveillance and knowledge of appropriate lifestyle habits were investigated using a self-administered questionnaire (in accordance with that proposed by the Ministry of Health, Labor and Human Welfare, Japan, with some modifications). Medical information on age, gender, height, current medical history, and medication status were obtained from clinical records, while weight, body fat percentage (BFP), BMI, and blood pressure (BP) were measured at the time of this study. Biochemical data on triglycerides, total cholesterol, high-density lipoprotein (HDL) cholesterol, low density lipoprotein (LDL) cholesterol, blood glucose, and HbA1c (as defined by the national glycohemoglobin standardization program of the USA) for suspected diabetes or diabetic patients were obtained from clinical blood laboratory tests. Biochemical tests were analyzed at the Nagoya Clinical Laboratory (Nagoya, Japan). Body weight (kg) and BFP (%BF) were measured using a health meter with a body fat scale (BF-046, Tanita, Tokyo, Japan), and BP was measured using a digital automatic blood pressure monitor (HEM-7071, Omron, Kyoto, Japan).

The self-administered questionnaire was based on the HBM [20] and included the following three major concepts: (i) “Knowledge“ including “Perceived susceptibility,” “Perceived severity” and “Perceived benefits”, (ii) “Perceived barriers”, (iii) “Cue to action”, (iv) “Self-efficacy”, (v) “Environment”, and finally (vi) “Satisfaction”. Our program was thus designed to incorporate these independent concepts. Participants were asked to choose the most applicable answer among the two to six grades for each questionnaire. For patients who were unable to answer these questionnaires by themselves, the conductor of the interview helped them to complete the form. These questionnaires were then scored.

### 2.3. Program Contents

The program content and objectives are listed in Table 1. The program utilized in this study is consisted of four steps: lecture (30 min), group work (10 min), goal setting and review (10 min), and questions and consultation (10 min). Eight lectures were held twice a month for 4 months, each consisting of basic principles such as “what is health?”, “how to control eating habits for snacks”, “the healthy lifestyle habits of thin people,”, “what is well-balanced diet?”, “the shopping knack to maintain good health”, “benefits of physical exercise”, “exercise practice” and the “review of these lectures”. Three of these eight sessions were conducted in a practical format, combining lectures with buffet meals and exercise. In the group work session following the lectures, questions related to what participants had learned in the lecture were asked in order to check their comprehension and to further participants’ understanding. Participants were encouraged to speak up in their own words. In the “Goal Setting” session that followed, participants discussed the changes in their lifestyle habits in relation to their previous goals, reviewed their goals, and set new goals. Finally, there was a “Question and Consultation time,” during which participants had time to ask questions of the instructor and discuss their individual problems and additional goals. The program consisted of eight sessions as described above, and for those who were unable to attend, a professional dietitian gave further lectures individually.

The HBM consisted of six components: “Perceived susceptibility”, “Perceived severity”, “Perceived benefits”, “Perceived barriers”, “Cue to action” and “Self-efficacy”. The first three components, “Perceived susceptibility”, “perceived severity” and “perceived benefits”, were made to understand the risks of getting lifestyle-related diseases. Additionally, the disadvantages of getting lifestyle-related diseases, and the benefits of being healthy through lectures and group work were consolidated through these activities.

The participants were encouraged to continue their progress and given advice on how to achieve their individual goals. In addition, in order to increase self-efficacy, each participant was asked to set a small goal at each meeting and to monitor their progress. The purpose of this goal-setting was to focus on the progress of each participant and to give each participant confidence in implementing the plan. If the program was perceived as difficult to complete, any participant could use the question/consultation time to overcome “perceived barriers” and resolve individual problems.

### 2.4. Control Group

Individuals enrolled in the control group were not given any program or other intervention. After the completion of the study, the same program was implemented for those who wished to participate.

### 2.5. Statistical Methods

The comparison between the intervention group and the control group was analyzed by a Mann–Whitney U test. Statistical analysis was performed using IBM SPSS statistics 26 (IBM Japan, Ltd., Tokyo, Japan) with a significance level of 5% (two-tailed test).

### 2.6. Study Registration and Ethical Approval

All subjects gave their informed consent for inclusion before they participated in the study. The study was conducted in accordance with the Declaration of Helsinki. This study was conducted after obtaining approval from the Ethical Review Committee for Human Subjects of Japan Women’s University (approval date: 8 August 2018, No. 337) and the Ethical Review Committee of Yahagigawa Hospital (approval date: 14 August 2018, No. 2018-005). In addition, the study was registered with the Japan Medical Association Center for Clinical Trial Promotion [JMACCT] (ID: JMA-IIA00365).

## 3. Results

### 3.1. Selection of Subjects

Of the 215 eligible participants, 185 were deemed ineligible through screening because they did not meet the requirements for enrollment, such as acute illness or lack of evaluation (180 cases), or because they refused to participate (5 cases). Thus, 30 cases were chosen as appropriate for the present study. These subjects were divided into two groups, the intervention and control groups, excluding any bias but matched for sex, age, and BMI (Figure 1).

### 3.2. Analysis of the Study Subjects

Demographic data on the participants are shown in Table 2. The control group consisted of 10 subjects (6 males and 4 females) and the intervention group consisted of 15 subjects (9 males and 6 females) with a mean age (standard deviation) of 60.4 (12.7) and 56.8 (12.1) years, respectively. These diagnoses of the patients in the control group were uniformly SZ, whereas the intervention group consisted of 12 patients (80%) with SZ and 3 with (20%) mood disorders. All were Japanese (100%).

In the intervention group, 14 (93.3%) completed all eight sessions of the program, and one person who could not complete the program withdrew from participation in the program immediately before the first session, but participated in a survey using a self-administered questionnaire administered after the completion of the eight sessions. In addition, one participant in the intervention group had polydipsia as a symptom of their condition, but completed the eight sessions of the program and participated in the two surveys. One person in the midway discharge (control group: *n* = 1) did not complete the final evaluation. Finally, 24 participants (24/25: 15 in the intervention group and 9 in the control group) completed the final evaluation.

### 3.3. Pre-Intervention Comparisons

Categorical variables were analyzed using Fisher’s exact test, while continuous variables were analyzed using Student’s t-test. There was a significant difference in the self-administered questionnaire items such as “Sweaty exercise for at least 30 min at least 2 days a week”, “Don’t you have any financial barrier?”, “Walking faster than others of the same age” (Table 3 and Table 4). However, these results appeared as not significant; thus, we did not pursue them any further.

### 3.4. Post-Intervention Comparison (Physical Status and Biochemical Tests)

As shown in Table 5, the test group who received the lifestyle modification program intervention showed a significant improvement in HDL cholesterol compared to the control group (*p* < 0.05) as analyzed via the Mann–Whitney U test. There was no significant difference in “body weight,” “BMI,” “BFP,” “BP,” “triglycerides,” “LDL cholesterol,” “blood glucose,”, and “HbA1c” values. The post power of analysis was 0.72 based on the results for HDL cholesterol, which was significantly different.

The values of the self-administered questionnaire were tested with the Mann–Whitney U test and found to be not significantly different. However, the number of those who understood the program content actually increased, though not statistically significantly (Table 6).

## 4. Discussion

In this pilot study we have examined the feasibility of a lifestyle modification program and found that there were significant effects on upregulating the plasma level of HDL cholesterol. Since we do not currently know of an effective medication to increase the level of HDL cholesterol, although there are a number of effective reagents to LDL cholesterol [31], our finding should give an interesting view for the treatment and prevention of hypercholesterolemia and, thus, of atherosclerosis. Further studies are needed to scale up this intervention program and increase the observation period. Another issue is whether the efficacy of this program is applicable to major psychiatric disorders. If the efficacy of this program is proved to be effective in general, it would be a significant development in the understanding and the treatment of hypercholesterolemia and atherosclerosis.

The HBM is a social model that attempts to assist subjects in perceiving the threat of disease and the effects of recommended health behaviors, in order to encourage them to improve their health. Panahi et al. [32] conducted a study of behavior promotion for osteoporosis prevention based on HBM and found significant improvements in preventive walking behaviors and knowledge. They attributed this to educational interventions including Q&A and encouraging messages. Differences from the current study include the fact that the study by Panahi et al. [32] was conducted with healthy subjects rather than patients, and incorporated the use of e-learning via the internet. Similarly, both studies included sufficient time for questions and focused on increasing self-efficacy. It should be noted that through these efforts, knowledge of disease prevention increased, and the time and frequency of walking exercise increased spontaneously. In general, increased knowledge directly led to action. Moreover, in this study, there was a significant improvement in HDL cholesterol levels, which cannot be achieved with ordinary dyslipidemia medications. On the other hand, the Nooriani et al. [22] investigated the impact of HBM-based interventions on the increase in nutritional knowledge regarding dietary intake in hemodialysis patients, but found no significant improvement in actual dietary intake; the study by Nooriani et al. included only outpatients, whereas the present study included inpatients. It is possible that snacking and exercise were easier to manage due to the participation and monitoring by medical staff involved in lifestyle and hospitalization facilitates during the nutritional education.

### Limitations

Regarding the duration of intervention, this study took place over 4 months. Continuous support should have given for a longer time in order to encourage active health behaviors among the participants. However, such continuation might not always be possible for patients with major psychiatric diseases. 

In addition, this study was conducted in a single hospital and had a small number of subjects. Although we performed this study by randomly assigning the intervention and the control groups as a pilot study, future studies with a larger study group should consider thorough random assignment to increase the statistical power.

It is worthwhile to conduct future research focusing not only on physical aspects but also on quality of life in the long run. In addition, the effectiveness of this program is not limited to those with mental illness, but may be effective for people in general. 

## 5. Conclusions

This program was based on the health belief model and is considered feasible for patients with psychiatric disorders, with obesity, and with lifestyle-related diseases. We would like to stress the observation that the current program was effective in improving biochemical risks for atherosclerosis to prevent lifestyle-related diseases. Future issues include extending the period of support, expanding the target population, and structuring program content, independent of the skills of the dietitian/practitioner. It is hoped that this study will lead to improvements in health conditions of a wide variety of patients and even healthy subjects.

## Figures and Tables

**Figure 1 healthcare-11-01690-f001:**
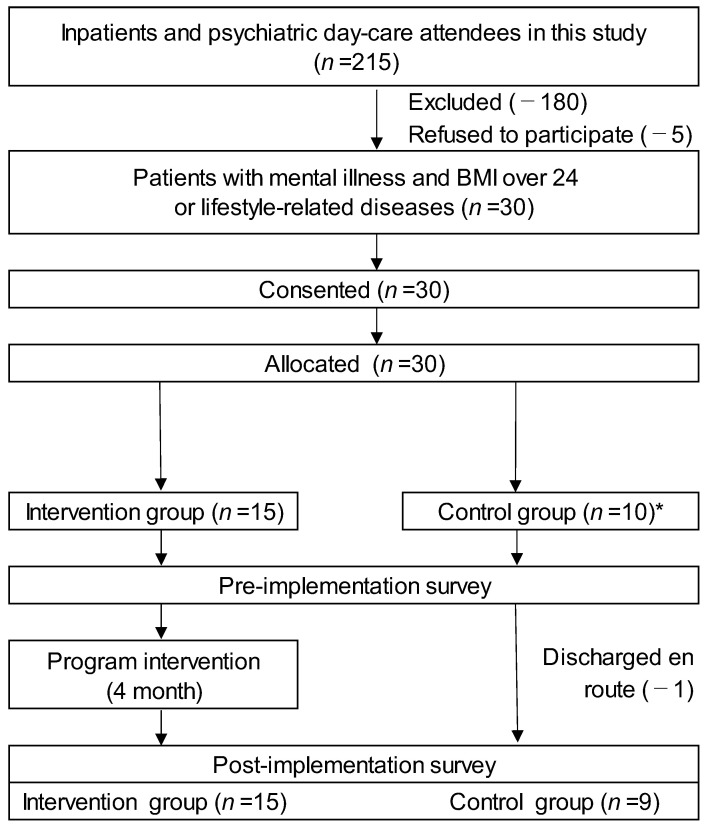
Flowchart of this study. * From the control group, 5 subjects voluntarily left the study knowing that they could not access the program.

**Table 1 healthcare-11-01690-t001:** Lifestyle improvement program content and aims.

Title	Aim of the Program	Lecture Content and Methods	Lecture Details	Group Work Questions
What is health?	Knowing what good health consists of and how to live a healthier life. Developing the ability to set goals.	The importance of health. (lecture)	Relationship between disease and lifestyle, showing pictures of diabetic gangrene.	What is a healthy lifestyle? What do you care about and what do you want to take care of regarding your health?
How to control eating habits for snacks	Understanding the advantages and disadvantages of snacking. Understanding what constitutes appropriate snacking. Developing the ability to change goals and continue.	The proper amount and selection of snacks. (lecture and practice)	Advantages and disadvantages of eating snacks (using actual snacks to demonstrate the proper amount.)	What are the advantages and disadvantages of eating sweets. What are the precautions in choosing snacks?
Healthy lifestyle habits of the healthy people	Understanding healthy lifestyle habits and the benefits of living a healthy lifestyle. Developing the ability to change goals and continue.	Healthy lifestyle habits. (lecture)	Healthy lifestyle habits and their advantages.	What do you need to pay attention to in your lifestyle?
What is the well-balanced diet?	Understanding the dietary balance and benefits of eating a balanced diet. Understanding what is an appropriate amount of food. Developing the ability to change and maintain the goal.	Eating a well-balanced diet (lecture and practice)	Well-balanced diet, followed by a buffet lunch.	What do you pay attention to regarding a well-balanced diet?
Shopping knacks to maintain good health	Knowing how to choose healthy products. Developing the ability to change and maintain the goal.	How to choose food for improved health. (lecture)	Choosing foods for health, correcting eating/shopping habits.	What do you buy when you are shopping? What do you need to pay attention to when you are shopping?
Benefits of physical exercise	Understanding the benefits of exercise. Knowing that exercise is easy to do. Developing the ability to change and maintain goals.	The Benefits of exercise (lecture)	Advantages of exercising and disadvantages of not exercising, simple ways to exercise.	What are some good/bad consequences of exercising or not exercising? What are some practical approaches to exercise?
Exercise practice	Knowing simple exercises. Developing the ability to change and maintain goals	Exercises that can be done at home (lecture and practice)	Review of the six sessions above. Practicing simple exercises that can be done at home.	What are your impressions after doing the exercise?
Review of these lectures/exercises	Recalling the previous content.	Review of the previous topics (lecture)	Review of the first seven sessions.	How well do participants comprehend the program after the 7th session? What has been learned, changed, or enabled through the program?

**Table 2 healthcare-11-01690-t002:** Participant characteristics and the number of program completers.

		Intervention Group (*n* = 15)	Control Group (*n* = 10)
Medical conditions	Psychiatric day care	6 (40%)	5 (50%)
	In the hospital	9 (60%)	5 (50%)
Gender	Male	9 (60%)	6 (60%)
	Female	6 (40%)	4 (40%)
Disease	Schizophrenia	12 (80%)	10 (100%)
	Mood disorder	3 (20%)	0
Mean	Age (years)	56.8 (12.1)	60.4 (12.7)
(Standard deviation)	BMI (kg/m^2^)	27.7 (3.7)	25.1 (3.4)
Program completers		14 (93.3%)	-

**Table 3 healthcare-11-01690-t003:** Pre-intervention comparisons (physical status, blood biochemistry).

Items (Average, SD)			Intervention Group (*n* = 15)	Control Group (*n* = 10)	*p*-Value
Physical status	body weight	(kg)	71.9 (13.9)	61.6 (9.4)	ns *
	BMI	(kg/m²)	27.7 (3.8)	25.1 (3.6)	ns
	body fat percentage	(%)	31.8 (7.3)	27.9 (7.8)	ns
	systolic BP	(mm Hg)	123 (21.3)	112 (21.1)	ns
	diastolic BP	(mm Hg)	79.6 (10.3)	73.2 (10.1)	ns
Blood biochemistry	blood sugar	(g/dL)	96 (33.5)	116 (48.5)	ns
	total cholesterol	(mg/dL)	183 (36.9)	172 (25.8)	ns
	triglyceride	(mg/dL)	182 (106)	134 (71.5)	ns
	HDL cholesterol	(mg/dL)	44.7 (10.4)	52.1 (13.3)	ns
	LDL cholesterol **	(mg/dL)	106 (30.4)	101 (35.7)	ns

* ns, not significant ** LDL cholesterol was calculated by formula F.

**Table 4 healthcare-11-01690-t004:** Pre-intervention comparisons (self-administered questionnaire).

Questions (*n*, %)		Options	Intervention Group (*n* = 15)	Control Group (*n* = 10)	*p*-Value
Knowledge about	Do you know what “health” is?	Not at all	2 (13.3)	2 (20)	ns
		I don’t think so.	3 (20)	0 (0)	
		Somewhat agree	0 (0)	2 (20)	
		Slightly agree.	6 (40)	2 (20)	
		I agree.	1 (6.7)	2 (20)	
		I agree very much.	3 (20)	2 (20)	
	Are you taking “healthy lifestyle”	Not at all	3 (20)	1 (10)	ns
		I don’t think so.	1 (6.7)	0 (0)	
		Somewhat agree	2 (13.3)	1 (10)	
		Slightly agree.	2 (13.3)	5(50)	
		I agree.	4 (26.7)	1(10)	
		I agree very much.	3 (20)	2(20)	
	Is your daily Diet well balanced?	Not at all	3 (20)	1(10)	ns
		I don’t think so.	0 (0)	0(0)	
		Somewhat agree	0 (0)	3 (30)	
		Slightly agree.	7 (46.7)	3 (30)	
		I agree.	2 (13.3)	1 (10)	
		I agree very much.	3 (20)	2 (20)	
	Do you know how to choose healthy Snacks?	Not at all	5 (33.3)	3 (30)	ns
		I don’t think so.	2 (13.3)	1 (10)	
		Somewhat agree	3 (20)	0 (0)	
		Slightly agree.	2 (13.3)	4 (40)	
		I agree.	2 (13.3)	1 (10)	
		I agree very much.	1 (6.7)	1 (10)	
	Do you go out for Shopping to buy healthy promoting goods?	Not at all	3 (20)	2 (20)	ns
		I don’t think so.	3 (20)	1 (10)	
		Somewhat agree	3 (20)	2 (20)	
		Slightly agree.	3 (20)	4 (40)	
		I agree.	1 (6.7)	1 (10)	
		I agree very much.	2 (13.3)	0 (0)	
	Do you know why Exercise is important?	Not at all	3 (20)	2(20)	ns
		I don’t think so.	1 (6.7)	1 (10)	
		Somewhat agree	1 (6.7)	2 (20)	
		Slightly agree.	5 (33.3)	5 (50)	
		I agree.	2 (13.3)	0 (0)	
		I agree very much.	3 (20)	0 (0)	
	Do you Exercise regularly?	Not at all	3 (20)	4 (40)	ns
		I don’t think so.	0 (0)	0 (0)	
		Somewhat agree	3 (20)	0 (0)	
		Slightly agree.	1 (6.7)	3 (30)	
		I agree.	3 (20)	2 (20)	
		I agree very much.	5 (33.3)	1 (10)	
	Do you know the appropriate Food intake?	Not at all	1 (6.7)	4 (40)	ns
		I don’t think so.	2 (13.3)	0 (0)	
		Somewhat agree	3 (20)	1 (10)	
		Slightly agree.	3 (20)	3 (30)	
		I agree.	3 (20)	1 (10)	
		I agree very much.	3 (20)	1 (10)	
Self-efficacy	Can you achieve the goals you expected?	Not at all	4 (26.7)	2 (20)	ns
		I don’t think so.	2 (13.3)	1 (10)	
		Somewhat agree	3 (20)	3 (30)	
		Slightly agree.	2 (13.3)	1 (10)	
		I agree.	1 (6.7)	2 (20)	
		I agree very much.	3 (20)	1 (10)	
	Can you achieve goals even when you are exhausted?	Not at all	4 (26.7)	3 (30)	ns
		I don’t think so.	3 (20)	1 (10)	
		Somewhat agree	2 (13.3)	3 (30)	
		Slightly agree.	2 (13.3)	2 (20)	
		I agree.	1 (6.7)	0 (0)	
		I agree very much.	3 (20)	1 (10)	
Perceived barriers	Do you feel difficulty in feeling healthy? (%)	I agree very much.	1 (6.7)	2 (20)	ns
		I agree.	0 (0)	2 (20)	
		Slightly agree.	8 (53.3)	2 (20)	
		Somewhat agree	0 (0)	1 (10)	
		I don’t think so.	2 (13.3)	1 (10)	
		Not at all	4 (26.7)	2 (20)	
Cue to action	Can you control your own diet?	Not at all	4 (26.7)	3 (30)	ns
		I don’t think so.	1 (6.7)	0 (0)	
		Somewhat agree	2 (13.3)	2 (20)	
		Slightly agree.	6 (40)	3 (30)	
		I agree.	0 (0)	1 (10)	
		I agree very much.	2 (13.3)	1 (10)	
	Can you control your lifestyle?	Not at all	2 (13.3)	5 (50)	ns
		I don’t think so.	1 (6.7)	1 (10)	
		Somewhat agree	4 (26.7)	1 (10)	
		Slightly agree.	3 (20)	2 (20)	
		I agree.	1 (6.7)	0 (0)	
		I agree very much.	4 (26.7)	1 (10)	
	Frequency of taking snacks	≥3/week	6 (40)	3 (30)	ns
		<3/week	9 (60)	7 (70)	
	Times of meals per day (within 2 h of bedtime)	≥3 times/week	2 (13.3)	1 (10)	ns
		<3/week	13 (86.7)	9 (90)	
	Frequency of taking breakfast	<4 times/week	1 (6.7)	1 (10)	ns
		≥4/week	14 (93.3)	9 (90)	
	Eating fast as compared to others	Slow	5 (33.3)	2 (20)	ns
		Normal	5 (33.3)	5 (50)	
		fast	5 (33.3)	3 (30)	
	Exercise	no	7 (46.7)	6 (60)	ns
		1~2 times/week	1 (6.7)	0 (0)	
		3~4 times/week	6 (40)	2 (20)	
		every day	1 (6.7)	2 (20)	
	Walking faster than others of the same age	Slow	4 (26.7)	5 (50)	*p* < 0.05
		Normal	4 (26.7)	5 (50)	
		fast	7 (46.7)	0 (0)	
	Sweaty exercise for at least 30 min at least 2 days a week	no	8 (53.3)	10 (100)	*p* < 0.05
		yes	7 (46.7)	0 (0)	
	Exercise for 1 h per day	no	9 (60)	10 (100)	ns
		yes	6 (40)	0 (0)	
Environment	Don’t you have any financial barrier?	I agree very much.	1 (6.7)	1 (10)	*p* < 0.05
		I agree.	0 (0)	2 (20)	
		Slightly agree.	1 (6.7)	4 (40)	
		Somewhat agree	3 (20)	1 (10)	
		I don’t think so.	0 (0)	1 (10)	
		Not at all	10 (66.7)	1 (10)	
	Do you have family support for health promotion?	Not at all	3 (20)	5 (50)	ns
		I don’t think so.	1 (6.7)	0 (0)	
		Somewhat agree	4 (26.7)	0 (0)	
		Slightly agree.	1 (6.7)	3 (30)	
		I agree.	2 (13.3)	0 (0)	
		I agree very much.	4 (26.7)	2 (20)	
	Do you have supportive environments for your health promotion?	Not at all	5 (33.3)	1 (10)	ns
		I don’t think so.	1 (6.7)	2 (20)	
		Somewhat agree	3 (20)	1 (10)	
		Slightly agree.	0 (0)	2 (20)	
		I agree.	3 (20)	2 (20)	
		I agree very much.	3 (20)	2 (20)	

**Table 5 healthcare-11-01690-t005:** Post-intervention comparisons (physical status, blood biochemistry).

Items			Before	After	Difference before and after	*p*-Value
			Average, (SD)	Average, (SD)	Average, (SD)	
Systolic BP	(mm Hg)	Intervention group (*n* = 15)	79.6 (10.3)	80.6 (9.5)	1 (9.7)	ns
		Control group (*n* = 9)	73.7 (10.5)	67 (7.6)	−6.7 (11.2)	
Diastolic BP	(mm Hg)	Intervention group (*n* = 15)	123 (21.3)	128 (23.0)	5.9 (16.4)	ns
		Control group (*n* = 9)	114 (21.2)	116 (16.8)	1.7 (24.2)	
BMI	(kg/m²)	Intervention group (*n* = 15)	27.7 (3.8)	27.3 (4.1)	−0.4 (0.6)	ns
		Control group (*n* = 9)	25.4 (3.7)	25.2 (3.8)	−0.1 (0.7)	
Body weight	(kg)	Intervention group (*n* = 15)	71.9 (13.8)	70.8 (14.0)	−1.0 (1.8)	ns
		Control group (*n* = 9)	62.3 (9.7)	61.9 (10.7)	−0.3 (1.9)	
Blood sugar	(g/dL)	Intervention group (*n* = 15)	96 (33.5)	104 (53.1)	8.9 (32.5)	ns
		Control group *(n* = 8)	122 (48.5)	99.7 (36.7)	−22.2 (56.3)	
Total cholesterol	(mg/dL)	Intervention group (*n* = 15)	183 (36.9)	193 (30.2)	10.4 (23.5)	ns
		Control group (*n* = 8)	171 (27.4)	168 (22.1)	−3.3 (19.2)	
Triglyceride	(mg/dL)	Intervention group (*n* = 15)	182 (106)	175 (153)	−6.8 (95.5)	ns
		Control group (*n* = 8)	142 (71.4)	133 (57.6)	−9.2 (62.7)	
HDL cholesterol	(mg/dL)	Intervention group (*n* = 15)	44.7 (10.4)	51.5 (11.7)	6.8 (7.6)	*p* < 0.05
		Control group (*n* = 8)	51.2 (13.9)	48.3 (12.2)	−2.9 (6.9)	
LDL cholesterol *	(mg/dL)	Intervention group (*n* = 14)	106 (30.4)	112 (28.5)	5.9 (16.9)	ns
		Control group (*n* = 8)	100 (38.1)	93.2 (21.9)	−7.7 (21.3)	

* LDL cholesterol was calculated by formula F.3.5. Post-Intervention Comparison (Self-Administered Questionnaire).

**Table 6 healthcare-11-01690-t006:** Changes in score post-intervention comparisons (self-administered questionnaire).

Items			Before	After	Difference before and after	*p*-Value
			Average	Average	Average	
Knowledge about	Do you know what “health” is?	Intervention group (*n* = 15)	4	5	0	ns
		Control group (*n* = 9)	4	3	0	
	Are you taking “healthy lifestyle”	Intervention group (*n* = 15)	4	4	0	
		Control group (*n* = 9)	4	5	1	
	Is your daily Diet well balanced?	Intervention group (*n* = 15)	4	5	0	ns
		Control group (*n* = 9)	4	4	0	
	Do you know how to choose healthy Snacks?	Intervention group (*n* = 15)	3	6	2	
		Control group (*n* = 9)	4	4	1	ns
	Do you go out for Shopping to buy healthy promoting goods?	Intervention group (*n* = 15)	3	5	1	
		Control group (*n* = 9)	3	4	1	
	Do you know why Exercise is important?	Intervention group (*n* = 15)	4	5	1	ns
		Control group (*n* = 9)	3	3	0	
	Do you Exercise regularly?	Intervention group (*n* = 15)	5	5	1	ns
		Control group (*n* = 9)	4	4	1	
	Do you know the appropriate Food intake?	Intervention group (*n* = 15)	4	5	1	ns
		Control group (*n* = 9)	3	4	2	
Self-efficacy	Can you achieve the goals you expected?	Intervention group (*n* = 15)	3	5	1	ns
		Control group (*n* = 9)	3	3	0	
	Can you achieve goals even when you are exhausted?	Intervention group (*n* = 15)	3	3	−1	ns
		Control group (*n* = 9)	3	3	0	
Perceived barriers	Do you feel difficulty in feeling healthy?	Intervention group (*n* = 15)	3	3	0	ns
		Control group (*n* = 9)	3	2	0	
Cue to action	Can you control your own diet?	Intervention group (*n* = 15)	4	4	1	ns
		Control group (*n* = 9)	3	4	1	
	Can you control your lifestyle?	Intervention group (*n* = 15)	4	5	1	ns
		Control group (*n* = 9)	1	4	1	
	Frequency of taking snacks	Intervention group *(n* = 15)	2	2	0	ns
		Control group (*n* = 9)	2	2	0	
	Times of meals per day (within 2 h of bedtime)	Intervention group (*n* = 15)	2	2	0	ns
		Control group (*n* = 9)	2	2	0	
	Frequency of taking breakfast	Intervention group (*n* = 15)	2	2	0	ns
		Control group (*n* = 9)	2	2	0	
	Eating fast as compared to others	Intervention group (*n* = 15)	2	2	0	ns
		Control group (*n* = 9)	2	2	0	
	Exercise	Intervention group *(n* = 15)	2	2	0	ns
		Control group (*n* = 9)	1	1	0	
	Walking faster than others of the same age	Intervention group (*n* = 15)	2	2	0	ns
		Control group (*n* = 9)	1	2	0	
	Sweaty exercise for at least 30 min at least 2 days a week	Intervention group (*n* = 15)	1	1	0	ns
		Control group (*n* = 9)	1	1	0	
	Exercise for 1 h per day	Intervention group (*n* = 15)	1	1	0	ns
		Control group (*n* = 9)	1	1	0	
Environment	Don’t you have any financial barrier?	Intervention group (*n* = 15)	6	3	−1	ns
		Control group (*n* = 9)	3	3	0	
	Do you have family support for health promotion?	Intervention group (*n* = 15)	3	5	0	ns
		Control group (*n* = 9)	1	1	0	
	Do you have supportive environments for your health promotion?	Intervention group (*n* = 15)	3	5	1	ns
		Control group (*n* = 9)	4	5	0	

## Data Availability

The data obtained will be available in an anonymized format so that each individual cannot be identified. Data requests must be made to the author.

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
