# Peer review of "Pilot and Feasibility Studies of a Lifestyle Modification Program Based on the Health Belief Model to Prevent the Lifestyle-Related Diseases in Patients with Mental Illness"

_healthcare, 2023, doi:10.3390/healthcare11121690_

Round 1

Reviewer 1 Report

Thank you.  The authors have examined the feasibility of a program based on the Health Belief Model for its effectiveness in improving lifestyle-related diseases in patients with SZ and MDI.  These program was very interested and I can agree this based intervention could be useful for general population. I wish the authors should have more patients with SZ or MDI and should analyze more experiments. However, this paper should be appeared scientific society and be useful in psychiatry area

Author Response

Thank you for kind and intensive review. We are glad that you gave us favorable comments. Since the other two reviewers gave us some helpful comments, we have amended our manuscript accordingly.

Reviewer 2 Report

Please explain median values in intervention and control groups in Table 5.

Author Response

Thank you very much for your insightful suggestion and positive comments.

Our response to Reviewer 2:

Criticism 1: Please explain median values in intervention and control groups in Table 5.

Our response: As mentioned by the Reviewer 3 (Criticism 12), we have deleted the median value from the figure.

Reviewer 3 Report

Dear Authors,

Thank you very much for the opportunity to read your manuscript. It is very interesting, but in my opinion it would require some changes:
- the abstract mentions 25 people, then there is information about 24 people who were alocated in the experimental and control groups
- there is a double space before: "These findings support "
- I suggest changing the MDI to bipolar disorder
- there is a problem with the numbering of sources - "1" are in superscripts
- these sentences are exclusive: "atients in acute care wards who were likely to be discharged from the hospital during the study period or who attended outpatient rehabilitation irregularly were excluded. These subjects were assigned to the study."
- what were the inclusion criteria? it should be described in more detail, because it will be the basis for further therapy in a specific group of patients: were the presence of active psychotic symptoms or cognitive impairment controlled? apart from HBM-based interventions, did the patients take part in other activities/therapy?
- if these are tools created by you (or modified), it is worth putting them in attachments
- it is worth preparing a chart showing subsequent measurements in the TAU and experimental group
- what does the power of analysis look like?
- for % in table 2, I suggest not adding 0 after the decimal point - all values are integers
- line 180 - I assume that this person has not been included in the analyses: this needs to be clarified
- 3.3 these results should be presented: e.g. in an attachment. for continuous variables, chi2 was used? if so - this should be changed to the appropriate calculation method
- there is an error in the calculation: U is used to compare two groups - your study is longitudinal, a different calculation method should be used
- since this is a pilot study, this information should be included in the title of the manuscript
- the calculations are wrong - this part of the manuscript should be corrected and the discussion changed accordingly.

Author Response

Author Reply to Reviewer 3

Thank you very much for your insightful suggestion and positive comments.

Our sentence-to-sentence responses to each criticism are the following:

Our response to Reviewer 3

1) Criticism 1: The abstract mentions 25 people, then there is information about 24 people who were allocated in the experimental and control groups

Our response: Our initial subjects for the control group were 25. However, one patient had discharged soon after the initiation of this study. Thus, the total number the control group was reduced from 10 to 9, which is described in the original manuscript (p.6, line 180-183).

2) Criticism 2: There is a double space before: "These findings support "

Our response: Agree. This extra-space was deleted in the revised manuscript.

3) Criticism 3: I suggest changing the MDI to bipolar disorder

Our response: Agree. We changed MDI to bipolar disorder (BD).

4) Criticism 4: There is a problem with the numbering of sources - "1" are in superscripts

Our response: Agree. We have modified as suggested.

5) Criticism 5: These sentences are exclusive: "patients in acute care wards who were likely to be discharged from the hospital during the study period or who attended outpatient rehabilitation irregularly were excluded. These subjects were assigned to the study."

Our response: Agree. We have revised the sentence as the following: “patients in acute care wards who were likely to be discharged from the hospital during the study period and those attended the outpatient rehabilitation irregularly were excluded.”

6) Criticism 6: What were the inclusion criteria? it should be described in more detail, because it will be the basis for further therapy in a specific group of patients: were the presence of active psychotic symptoms or cognitive impairment controlled? apart from HBM-based interventions, did the patients take part in other activities/therapy?

Our response: Agree and disagree. In the original manuscript we have mentioned the inclusion criteria such as exclusion of patients in active psychotic conditions quite concretely (page 2, lines 74-77). However, in the revised manuscript we have mentioned that these subjects were not included in other clinical studies. added more detailed criteria for the inclusion criteria (page 2, lines 78-79).

7) Criticism 7: If these are tools created by you (or modified), it is worth putting them in attachments

Our response: Agree and Disagree. This search program is entirely original and there is no premade tool or form. The details of study questionnaire and the method of analysis are described in this paper. 

8) Criticism 8: It is worth preparing a chart showing subsequent measurements in the TAU and experimental group

Our response: What is “TAU”? Anyhow, as described in this paper, we have opened every relevant data obtained. We do not understand what this reviewer really wants to say.

9) Criticism 9: What does the power of analysis look like?

Our response: We do not understand what this reviewer really wants to say. Our manuscript speaks itself.

10) Criticism 10: For % in table 2, I suggest not adding 0 after the decimal point - all values are integers

Our response: Agree.  We have corrected the manuscript Table 2 accordingly.

11) Criticism 11: Line 180 - I assume that this person has not been included in the analyses: this needs to be clarified

Our response: Disagree. We are afraid that this is a misunderstanding of this reviewer. This patient has been included in the analysis. However, as mentioned in the original manuscript, this patient was dropped but her data was available. This sometime happens in a study including psychic patients. 

12) Criticism 12: 3.3 these results should be presented: e.g. in an attachment. for continuous variables, chi2 was used? if so - this should be changed to the appropriate calculation method.

Our response: Agree. Actually, the Table 3 of the original manuscript happened to be taken from our old incomplete manuscript. Thus, this referee pointed out the ambiguity of the data. We are sorry about this and have replaced this table by the correct one. We have included the actual results in a new Table 3. Since the valuables are continuous, Fisher’s exact test and Student’s t-test were applied instead of chi-square test. Interestingly, we found some inspection items have shown significant differences among intervention and non-intervention groups. However, these results are not mentioned in the text because of the irrelevance in the context.

13) Criticism 13: There is an error in the calculation: U is used to compare two groups - your study is longitudinal, a different calculation method should be used

Our response: Agree but the results disagree your view. We used Mann-Whitney test because the change of each items has no reason to follow normal distribution. We also tried to analyze the results by Student’s t-test. However, the result with our current observatory data did show essentially the same result.  

14) Criticism 14: Since this is a pilot study, this information should be included in the title of the manuscript.

Our response: Agree. We admit this comment and the title has been modified accordingly.

15) Criticism 15: The calculations are wrong - this part of the manuscript should be corrected and the discussion changed accordingly.

Our response: We have responded this comment in the above (see our response to Criticism 12,13).
